# 1 Methane quantification of LNG gas-fired power plant in Seoul, South Korea

2 Jaewon Joo<sup>1,3</sup>, Sujong Jeong<sup>2,3\*</sup>, Hyukjae Lee<sup>2</sup>, Yeonsoo Kim<sup>2</sup>, Jaewon Shin<sup>2</sup>, Donghee Kim<sup>2</sup>, 3 Dongyoung Chang<sup>1,3</sup> 4 5 <sup>1</sup>Environmental Planning Institute, Seoul National University, Seoul, Republic of Korea <sup>2</sup>Department of Environmental Planning, Graduate school of Environmental Studies, Seoul 6 National University, Seoul 08826, Republic of Korea. 7 8 <sup>3</sup>Climate Tech Center, Seoul National University, Republic of Korea 9 \* Corresponding Author: Sujong Jeong, sujong@snu.ac.kr 10 11 12

## Abstract

14

15 Methane emissions from a liquefied natural gas (LNG) gas-fired power plant in Seoul, South Korea were measured using a mobile greenhouse gas measurement platform. Twenty-one 16 mobile measurements were conducted between February and July 12, 2023. Methane emissions 17 were quantified using the Gaussian Plume Dispersion Model and the OTM-33A method. The 18 19 measurements identified three key emission hotspots: two associated with natural gas pipelines (S1 and S2), and one linked to an exhaust pipe from internal facilities (S3). The average 20 methane emission rates were  $0.09 \pm 0.0086$ ,  $0.018 \pm 0.0015$ , and  $0.55 \pm 0.0583$  tons hr<sup>-1</sup> at S1, 21 S2, and S3, respectively. Notably, S3 had a significant methane emission rate of  $2.053 \pm 0.283$ 22 23 tons hr<sup>-1</sup>, approximately six times greater than our corresponding bottom-up estimate of fugitive methane emissions (0.35 tons hr-1). This significant discrepancy, particularly at S3, 24 25 highlights the limitations of bottom-up inventory approaches and underscores the importance of field measurements for accurately assessing real-world emissions. This study provides 26 27 crucial evidence that mobile measurements are useful in identifying and quantifying fugitive methane emissions from urban LNG power plants. These findings are essential for developing 28 a more precise understanding of effective methods to reduce methane emissions from these 29 facilities. 30

- Keywords: Mobile measurements, Fugitive methane emissions, Methane quantification, LNG
- power plant

#### 1. Introduction

The global demand for liquefied natural gas (LNG) reached 401 million tons in 2023, with market growth driven largely by Asia, which accounts for 64% of global LNG imports (Giignl, 2024; Union, 2024). China recently overtook Japan as the world's largest LNG importer, with imports reaching approximately 72 million tons, whereas Japan imported approximately 66 million tons in the same year (Union, 2024). South Korea remains among the world's top three importers, with imports of approximately 45 million tons in 2023 (Giignl, 2024; Union, 2024). Due to its relatively low carbon intensity compared to coal or oil, LNG is often recognized as a "transition fuel" for decarbonization efforts (Al-Kuwari, 2023; Union, 2024). The concept of LNG as a transition fuel has been supported by previous studies, highlighting its potential to facilitate a shift from more carbon-intensive sources and to support the intermittency of renewable energy (Al-Kuwari, 2023). However, this perspective is challenged by research showing that methane emissions associated with LNG supply chains and gas-fired power generation can significantly undermine this environmental benefit<sup>4</sup>, as methane has a much higher global warming potential than carbon dioxide over a shorter time period (IPCC, 2023).

In South Korea, the energy sector, including LNG gas-fired power plants, accounts for approximately 23% of total national methane emissions (MOE, 2022). The number of LNG gas-fired power plants in the country has expanded in recent years to meet rising electricity demands, with plans to increase LNG-based power capacity from 43.3 GW in 2020 to 69.5 GW by 2038 under the 11th National Basic Plan for Electricity Supply and Demand in South Korea. This transition offers a cleaner alternative to traditional coal-fired facilities in terms of carbon dioxide emissions. However, the accurate quantification of methane emissions from LNG power plants has proven challenging (Lyon et al., 2015). Most methane estimates for these facilities rely on bottom-up inventories, which often have difficulty capturing fugitive emissions from operating conditions (Alvarez et al., 2018; Brandt et al., 2014). These inventories, which apply generic emission factors to activity data, can underestimate actual emissions owing to their limitations in reflecting real-world variability (Howarth, 2024).

To address the limitations of traditional bottom-up inventories, which often struggle

69

82

85

93

to capture fugitive emissions from complex industrial operations, top-down measurement approaches utilizing mobile platforms have gained significant traction in recent years for quantifying urban methane emissions (IPCC, 2023; Alvarez et al., 2018; Brandt et al., 2014). Mobile measurements from vehicle-mounted sensors have proven to be highly effective for mapping and identifying fugitive methane emissions from urban natural gas distribution networks (Vogel et al., 2024). This provides a more accurate understanding of urban methane budgets. Mobile measurements identify more fugitive methane sources that are difficult to detect in bottom-up inventories and provide data for improved estimations of total emissions (Joo et al., 2024; Maazallahi et al., 2022; Maazallahi et al., 2020; Vogel et al., 2024; Mitchell et al., 2015; Jia et al., 2025). Vogel et al. (2024) conducted a study in 12 European cities and demonstrated that mobile measurements could effectively identify and quantify methane emissions from natural gas systems in diverse urban infrastructures. These measurements successfully detected methane emissions that traditional approaches have often missed. In Hamburg, Germany, Maazallahi et al. (2022) quantified urban natural gas emissions using a vehicle-based methane monitoring system and found that fugitive emissions accounted for approximately 15% of the total estimated emissions, with individual leakage rates varying from 0.1 to 5 kg hr<sup>-1</sup>. Joo et al. (2024) discovered significant missing fugitive methane emissions (approximately 573 tons per year) from urban sewer networks in the Gwanak district of Seoul, South Korea, highlighting the limitations of relying solely on bottom-up inventories. Beyond urban infrastructure, mobile platforms are crucial for assessing fugitive methane emissions from industrial facilities. Studies in Texas and California, U.S., have quantified fugitive emissions and identified major sources by comparing top-down methane measurements and bottom-up inventories around oil and gas production and processing sites (Alvarez et al., 2018; Brandt et al., 2014; Lyon et al., 2015). Alvarez et al. (2018) evaluated methane emissions across the U.S. oil and gas supply chain, estimating total emissions at 13 million tons per year, a figure 60% higher than the U.S. Environmental Protection Agency (EPA) estimates at the time. Furthermore, Brandt et al. (2016) quantified methane leaks in a North American natural gas system and reported leak rates ranging from 0.05% to 8% of the natural gas production, with substantial variation across different supply chain segments. Jia et al. (2025) quantified fugitive methane emissions from natural gas stations in China using on-site component-level measurements and identified "super-emitters," which accounted for nearly 80% of the total https://doi.org/10.5194/egusphere-2025-4379 Preprint. Discussion started: 27 October 2025 © Author(s) 2025. CC BY 4.0 License.

fugitive emissions detected at the LNG facilities, with methane concentrations exceeding 10,000 ppm. These studies highlighted the importance of direct measurement techniques for quantifying fugitive methane emissions and identifying key mitigation opportunities in the natural gas industry.

This study addresses the need for a comprehensive assessment of methane emissions from LNG gas-fired power plants, particularly those located within major urban infrastructures. This is one of the first studies to measure methane production at a major LNG power plant in metropolitan Seoul. Using a mobile greenhouse gas (GHG) measurement platform, we identified and quantified the fugitive methane sources within a large LNG gas-fired power plant. This study also compared top-down measurements with bottom-up inventory estimates of fugitive methane emissions. Using a mobile GHG measurement platform, this study aimed to provide a more accurate assessment of fugitive methane emissions, overcoming the limitations associated with bottom-up methods in real environments. A comparison between top-down measurements and bottom-up inventories will contribute to a better understanding of the gap between current emissions reporting and real environmental methane emissions of LNG-based power generation in urban areas.

**2. Method** 

## 2.1 Study area and mobile GHG platform

The target LNG gas-fired power plant (Figure 1) is one of the largest underground LNG power 114 plants (800 MW) in the world, generating approximately 86.7% of the total electricity in Seoul 115 (4,435 GWh in 2023), according to Seoul open data (Seoul Metropolitan Governments, 2025). 116 117 The target LNG gas-fired power plant is an underground facility, and an urban renewal park was created for public use. 118 119 The mobile GHG platform used in this study comprises an electric vehicle equipped with a 120 global positioning system (GPS) and GHG analyzers for measuring CO<sub>2</sub>, CH<sub>4</sub>, and C<sub>2</sub>H<sub>6</sub> concentrations, as shown in Figure 2 (Joo et al., 2024). The platform featured the LI-7810 121 122 analyzer, which employs optical feedback cavity-enhanced absorption spectroscopy to measure 123 atmospheric CH<sub>4</sub>, CO<sub>2</sub>, and H<sub>2</sub>O concentrations at 1-s intervals with a precision of  $\pm$  0.6 ppb 124 for CH<sub>4</sub>. Additionally, the GLA131-MEA analyzer used off-axis integrated cavity output spectroscopy to measure CH<sub>4</sub> and C<sub>2</sub>H<sub>6</sub> with a precision of ± 0.09 ppb and ± 20 ppb, 125 126 respectively. Both analyzers were calibrated using reference gas cylinders before mobile 127 measurements were conducted. The GPS system employed an AK-770 device that integrates GPS and GLONASS to provide accurate location data, including longitude, latitude, speed, 128 129 and elevation at 1-s intervals with a precision of  $\pm$  20 m. An electric vehicle (KIA EV6) was selected to eliminate combustion-related emissions during the measurements, and the analyzer 130 inlets were installed on the roof of the vehicle at a height of 2.1 m to conduct GHG sampling. 131

**Figure 1.** Target area for methane measurements (© OpenStreetMap contributors 2024, distributed under the Open Database License (ODbL 1.0). Tiles by Carto, licensed under CC BY 3.0; © Google Maps 2024)

Mobile GHG measurement platform

GLA131-MEA

Inlet

LI-7810

GLA131-MEA

AK-770

Figure 2. Mobile GHG measurement platform

### Table 1. Specifications of mobile GHG measurement platform

| Instruments    | Manufacturer | Туре                                                                                      | Time step               | Range                                                                                                                                      | Precision                                                                                                                                                             |
|----------------|--------------|-------------------------------------------------------------------------------------------|-------------------------|--------------------------------------------------------------------------------------------------------------------------------------------|-----------------------------------------------------------------------------------------------------------------------------------------------------------------------|
| LI-7810        | LICOR        | CH <sub>4</sub> , CO <sub>2</sub> , H <sub>2</sub> O                                      | 1 second                | CH <sub>4</sub> : 0 – 100 ppm<br>CO <sub>2</sub> : 0 – 10,000<br>ppm<br>H <sub>2</sub> O: 60,000 ppm                                       | $CH_4$ : $\pm 0.6$ ppb $CO_2$ : $\pm 3.5$ ppm $H_2O$ : $\pm 45$ ppm                                                                                                   |
| GLA131-<br>MEA | LGR          | CH4, C2H6, H2O                                                                            | 1 second (± 0.2 second) | CH <sub>4</sub> : 0 – 10,000<br>ppm<br>C <sub>2</sub> H <sub>6</sub> : 0 – 500 ppm                                                         | $CH_4:\pm 0.9 \text{ ppb}$ $C_2H_6\pm 20 \text{ ppb}$                                                                                                                 |
| WPSD-9100      | YOUNG        | Wind direction<br>(WD),<br>Wind speed (WS)                                                | 1 second                | WD: 0 – 359.9°<br>WS: 0 – 50 m s <sup>-1</sup> ,                                                                                           | $WD: \pm 1^{\circ} \\ WS: \pm 2^{\circ}RMSE \\ from 1 m s^{-1}$                                                                                                       |
| EOLOS-<br>IND  | LAMBRECHT    | Wind direction (WD) Wind speed (WS) Humidity (H) Temperature (T) barometric pressure (BP) | 1 second                | WD: $0 - 360^{\circ}$<br>WS: $0.1 - 85 \text{ m s}^{-1}$<br>H: $0 - 100 \%$<br>T: $-40 \sim 70 ^{\circ}$ C<br>BP: $600 - 1100 \text{ hpa}$ | WD: $\pm 3^{\circ}$<br>WS: $\pm 0.5 \text{ m s}^{-1} \pm 5\%$<br>H: $\pm 3\% \pm 4\%$<br>T: $0.8 ^{\circ}\text{C} \text{ (v > 2m s}^{-1})$<br>BP: $\pm 2 \text{ hpa}$ |
| AK-770         | ASCEN Korea  | GPS                                                                                       | 1 second                | Longitude,<br>Latitude, Speed                                                                                                              | ± 20m                                                                                                                                                                 |
| EV6            | KIA Motors   | Vehicle                                                                                   | -                       | -                                                                                                                                          | -                                                                                                                                                                     |

## 2.2 Quantification of methane emissions from top-down approaches

The methane concentration data from mobile measurements at the LNG gas-fired power plant were quantified using the standard point source Gaussian Plume Dispersion Model (GPDM) and Other Test Method 33A (OTM-33A) (see eqs. (1) and eqs. (2)). The GPDM is a widely used atmospheric dispersion model that assumes a Gaussian distribution of methane concentrations in the horizontal and vertical directions under steady-state conditions (Turner, 1970; Chen et al., 2020; Maazallahi et al., 2020). It considers factors such as the emission rate, wind speed, atmospheric stability, and distance from the source to estimate pollutant concentrations downwind. OTM-33A, developed by the U.S. EPA, is a near-source flux measurement method designed to locate and estimate methane emissions from oil and gas facilities without requiring site access (Thoma and Squier, 2014). It employs an inverse Gaussian approach, and is particularly useful for mobile measurements. Both methods offer advantages in capturing real-world emissions under operational conditions, and can help

quantify specific emission sources within a facility.

$$C(x, y, z) = \frac{Q}{2\pi u \sigma_y \sigma_z} \left\{ \exp\left(\frac{-(z - z_{source})^2}{2\sigma_z^2}\right) \exp\left(\frac{-(z + z_{source})^2}{2\sigma_z^2}\right) \right\} \exp\left(\frac{-y^2}{2\sigma_y^2}\right)$$
(1)

where C is the CH<sub>4</sub> enhancement converted to grams per cubic meter (g m-3) at Cartesian coordinates x, y, and z relative to the source ([xyz]source=0 at ground-level source); x is the distance of the plume from the source aligned with the wind direction; y is the horizontal axis perpendicular to the wind direction; z is the vertical axis; Q is the emission rate in grams per second (g s<sup>-1</sup>); u (m s<sup>-1</sup>) is the wind speed along the x axis; and  $\sigma_y$  and  $\sigma_z$  are the horizontal and vertical plume dispersion parameters, respectively.

$$Q = 2\pi \cdot \sigma_y \cdot \sigma_z \cdot U \cdot C \tag{2}$$

where Q (g s<sup>-1</sup>) is the source emission rate.  $\sigma_y$  and  $\sigma_z$  are the horizontal and vertical dispersion coefficients, respectively, from the Pasquill-Gifford stability class listed in Table 2. U is the average wind speed during the measurement (m s<sup>-1</sup>) (Thoma and Squier, 2014).

Table 2. Pasquill–Gifford stability class (Thoma and Squier, 2014)

|                                         | Day with insolation |          |        | Night                      |               |
|-----------------------------------------|---------------------|----------|--------|----------------------------|---------------|
| Surface wind speed (m s <sup>-1</sup> ) | Strong              | Moderate | Slight | Overcast or >4/8 Low cloud | <3/8<br>Cloud |
| < 2                                     | A                   | A~B      | В      | -                          | -             |
| 2~3                                     | A~B                 | В        | С      | Е                          | F             |
| 3~5                                     | В                   | B~C      | D      | D                          | Е             |
| 5~6                                     | С                   | C~D      | D      | D                          | D             |
| > 6                                     | С                   | D        | D      | D                          | D             |

176 A: Extremely unstable, B: Moderate unstable, C: Slightly unstable, D: Neutral, E: Slightly stable, E:

Moderately stable

#### 2.3 Quantification of fugitive methane emissions from bottom-up approaches

The hourly fugitive methane emission rates of the target LNG gas-fired power plant were estimated by multiplying the activity data with the emission factor.

$$Methane\ emission\ rate = Activity\ data\ \times Emission\ factor \tag{3}$$

The activity data used in this study were the monthly LNG consumption of the power plant reported by the KOREA MIDLAND POWER Co., Ltd. (KOMIPO). For the emission factor, South Korea's country-specific factor for fugitive emissions from post-meter leakage at industrial plants and power stations (87.5 tCH<sub>4</sub>/PJ<sup>6</sup>) was applied. The resulting monthly emissions were then distributed on an hourly basis using 5-min operational status data from the Korea Power Exchange as a proxy (KPX, 2025), which records the near real-time power generation estimates of the target plant in megawatts (MW).

### 2.4 Measurement strategy

LNG gas-fired power plants in South Korea are mostly restricted because of security issues; thus, indirect measurement strategies are required. The target LNG gas-fired power plant in this study is an underground power plant. Methane measurements were conducted in the underground and ground regions. In this study, we employed the GPDM and OTM-33A to quantify the methane emissions from an LNG gas-fired power plant. The measurement strategies for these two models were selected as mobile and stationary measurements. The mobile measurement strategy drives multiple trajectories along the boundaries of the target LNG power plant. In this study, we drove 10 to 15 trajectories to quantify methane emissions using the GPDM. The stationary measurement strategy involved at least 30 min of measurement near the identified methane emissions. To identify methane leaks from the LNG

https://doi.org/10.5194/egusphere-2025-4379 Preprint. Discussion started: 27 October 2025 © Author(s) 2025. CC BY 4.0 License.

- power plant, we conducted driving and walking monitoring for methane leak surveys in the
- LNG power plant area.

#### 3. Results and Discussion

Figure 3 illustrates three key methane emission hotspots (S1, S2, and S3) using a mobile GHG platform near an LNG gas-fired power plant. Repeated mobile transect and targeted walking surveys consistently detected elevated methane concentrations at these hotspots. Methane enhancements in indicate a maximum of 3,795.7 ppb (S1) with an average of 1,698.62 ppb, a maximum of 1,188.94 ppb (S2) with an average of 466.22 ppb, and a significant maximum of 56,039.06 ppb (S3) with an average of 19,963.97 ppb, confirming them as areas significantly impacted by emissions from the LNG gas-fired power plant. S1 and S2 were located downwind of sections of the plant's natural gas pipelines and LNG power plant facilities, such as power generation units and smokestacks. S3 was located downwind of an exhaust pipe associated with internal processes and LNG power plant facilities. The mobile measurement strategy employed in this study was effective in monitoring the target area surrounding the access-restricted LNG power plant, enabling the identification and characterization of methane emission plumes from the facilities.

The contrasting methane emission characteristics of S1, S2, and S3 are shown in Figs. 4 and 5. Figure 4 shows the relatively constant methane emissions during the 10–15 mobile measurement trajectories at S1 and S2. In contrast, Figure 5 highlights the frequent and pronounced spikes in the methane and ethane concentrations measured downwind of the exhaust source (S3). Methane and ethane concentration data can be used to distinguish between fossil fuels and microbial sources (Joo et al., 2024). Ethane is co-emitted with methane from fossil fuel sources, such as natural gas leaks, whereas microbial processes typically produce methane with negligible amounts of ethane (Maazallahi et al., 2022; Joo et al., 2024). Figure 5 shows that the ethane concentrations increase and decrease in a pattern similar to the methane concentrations at S3, strongly suggesting that the methane emissions measured at S3 originate from the LNG power plant.

Methane emission rates at S1 and S2 were quantified using the GPDM from the mobile measurement data. The average emission rates of these hotspots were  $0.09 \pm 0.0086$  tons hr<sup>-1</sup> for S1 and  $0.018 \pm 0.0015$  tons hr<sup>-1</sup> for S2 (Table 3 and Figure 4). The emission rate at hotspot S3 was quantified using OTM-33A, which is appropriate for the higher concentrations captured in this area, with an average emission rate of  $0.55 \pm 0.0583$  tons

238239

241242

244245

246247

249250

258259

262263

264265

hr<sup>-1</sup> (Table 3, Figure 5). Notably, the emissions at S3 exhibited substantial temporal fluctuation, ranging from  $0.064 \pm 0.009$  tons hr<sup>-1</sup> to  $2.053 \pm 0.283$  tons hr<sup>-1</sup> (May 31st; Table 3). Quantification of the methane emission rates at the three hotspots revealed distinct characteristics for the natural gas pipeline-associated locations versus the exhaust pipes of the LNG power plant-associated locations in Table 3 and Figs. 4 and 5. The two hotspots located downwind of the natural gas pipelines and LNG power plant facilities, S1 and S2, exhibited relatively lower and more consistent methane emission rates. In contrast, hotspot S3, which was located downwind of the exhaust pipe and LNG power plant facilities, displayed significantly higher average emissions and pronounced variability. This extreme variability at S3 strongly suggests the occurrence of intermittent, high-magnitude emission events potentially linked to specific operational phases (such as startups, shutdowns, or load changes) or process inefficiencies (such as incomplete combustion), aligning with the concept of 'super-emitter' behavior noted in other industrial emission studies (Alvarez et al., 2018; Brandt et al., 2014).

Table 3 shows the significant discrepancies observed between our emission quantifications derived from mobile measurements and fugitive methane estimates based on standard national emission factors and activity data from the target LNG power plant. Although bottom-up inventories generally underestimated emissions, particularly during the significant peak event at S3, there were other instances, particularly at S1 and S2, where the inventories overestimated the measured emissions. Specifically, bottom-up estimates of this study showed a relatively narrow range of 0.18–0.499 tons hr<sup>-1</sup>. In contrast, the measured emissions exhibited significant variability across sites, ranging from 0.007 to 0.302 tons hr<sup>-1</sup> at S1, 0.002 to 0.047 tons hr<sup>-1</sup> at S2, and 0.005 to 2.053 tons hr<sup>-1</sup> at S3. The maximum methane emission rate at S3 on May 31st, 15:00-16:00, was  $2.053 \pm 0.283$ tons hr<sup>-1</sup>, which was significantly higher than the bottom-up estimate of 0.35 tons hr<sup>-1</sup>. However, Table 3 also indicates that in several instances at S1 and S2, the bottom-up estimates were higher than the measured methane emissions. Discrepancies between fugitive methane estimates from bottom-up inventories and methane emissions from mobile measurements are influenced by several critical factors in LNG power plants. Operational variability, such as startups and shutdowns, can lead to short-term spikes in methane emissions that are difficult to capture using bottom-up methane estimates (Brandt

269270

276277

278279

280281

289290

293294

et al., 2014). Furthermore, undetected or slowly developing fugitive emissions from the LNG facilities such as pipelines and fittings can be identified as intermittent 'super-emitter' events by top-down measurements rather than bottom-up methane estimates (Howarth, 2019; Alvarez et al., 2018; Karion et al., 2013). This variability underscores the limitations of bottom-up approaches for accurately capturing real-world operational fluctuations.

These findings are consistent with previous research, highlighting the difficulties faced by bottom-up methods in fully accounting for fugitive emissions from LNG power plants (Alvarez et al., 2018; Brandt et al., 2014; Howarth, 2019; Karion et al., 2013; Zavala-Araiza et al., 2015). Alvarez et al. (2018) reported that the actual methane emissions from the U.S. oil and natural gas supply chain were approximately 60% higher than the estimates provided by EPA inventories. Similarly, our top-down measurements at the S3 hotspot revealed peak emissions of  $2.053 \pm 0.283$  tons hr<sup>-1</sup>, which are approximately six times greater than our corresponding bottom-up estimate of 0.35 tons hr<sup>-1</sup>. This level of discrepancy aligns with measurements of "super-emitter" phenomena in natural gas facilities. Mitchell et al. (2015) found that the top 30% of natural gas gathering facilities contributed 80% of the total emissions. One facility alone accounted for 10% of all measured emissions from the gathering facilities. This indicates that a few sources can have a disproportionate impact on the overall emissions. Mitchell et al. (2015) reported a median throughput-normalized weighted average facility-level emissions rate of 0.079% for processing plants. These plants differ from LNG power plants but have similar components. The magnitude of our peak S3 emission (2.053 tons hr<sup>-1</sup>) suggests a significant emission event that far exceeds typical operational estimates. This level of discrepancy aligns with "super-emitter" phenomena measurements in natural gas facilities. Jia et al. (2025) directly measured fugitive methane emissions at several natural gas facilities in China, including an LNG terminal. They found that components with methane concentrations exceeding 10,000 ppm, although constituting only approximately 10% of the leaking components, accounted for approximately 80% of the total methane emissions. Total fugitive methane emissions detected at the LNG terminal were approximately 0.59 tons hr<sup>-1</sup>. The overall emission magnitude at the LNG terminal in their study (0.59 tons hr  $^{1}$ ) is comparable to our average emission rate at S3 of 0.55  $\pm$  0.0583 tons hr $^{1}$  but significantly lower than our maximum methane emission rate at S3 from mobile

measurements. The discrepancy observed at S3 highlights the importance of top-down measurements for capturing highly intermittent emission events that bottom-up inventories may not accurately represent.

**Figure 3**. Key hotspots of methane emissions in the target area from the mobile measurements (© Google Maps 2024)

**Figure 4**. Methane emissions from the mobile measurements at S1 and S2 for the GPDM on 23 June 2023. (a) driving route with the two major methane sources from the LNG power plant (color scale); the yellow star mark shows the location of anemometer. (b) and (c) indicate CH<sub>4</sub> enhancement variabilities (ppb) from two sources versus distance along the route. Black dotted

points are bin-mean values from 15 trajectories; the red line represents a Gaussian fit to those points. Blue annotations indicate  $\sigma$  (lateral dispersion length scale, m) and the mean wind speed  $(u, \text{m s}^{-1})$  (© Google Maps 2024)

**Figure 5.** Methane emissions from mobile measurements at S3 for OTM-33A on 23 June 2023. (a) indicates the photograph of the stationary measurement point; red circles (inset) show the major methane source of LNG power plant (S3). (b) represents the centered wind direction time series with reference lines at  $0^{\circ}$  (red) and  $\pm 30^{\circ}$  (blue); the header shows the mean centered wind direction and the fraction within  $\pm 30^{\circ}$ . (c) represents the CH<sub>4</sub> and C<sub>2</sub>H<sub>6</sub> enhancements (ppb).

# 324 Table 3. Quantification of methane emissions by mobile measurements and bottom-up GHG

# 325 inventory estimates

| Date     | Time        | Methane emission rate (ton hr <sup>-1</sup> , 2023)   |                                                       |                                                       | GHG inventory                           |
|----------|-------------|-------------------------------------------------------|-------------------------------------------------------|-------------------------------------------------------|-----------------------------------------|
| Date     |             | Hotspot 1                                             | Hotspot 2                                             | Hotspot 3                                             | estimation (ton hr <sup>1</sup> , 2023) |
| 23.04.26 | 14:00~15:00 | -                                                     | -                                                     | $0.499 \pm 0.0682452$                                 | 0.19                                    |
| 23.05.03 | 14:00~15:00 | 1                                                     | -                                                     | $\begin{array}{c} 0.064 \pm \\ 0.0087912 \end{array}$ | 0.24                                    |
| 23.05.12 | 14:00~15:00 | -                                                     | -                                                     | $0.131 \pm 0.0182124$                                 | 0.18                                    |
| 23.05.19 | 9:00~12:00  | $\begin{array}{c} 0.302 \pm \\ 0.0416124 \end{array}$ | $0.047 \pm 0.006534$                                  | -                                                     | 0.35                                    |
| 23.05.26 | 09:00~10:00 | $0.069 \pm 0.0094896$                                 | $\begin{array}{c} 0.007 \pm \\ 0.0009108 \end{array}$ | -                                                     | 0.32                                    |
|          | 10:00~11:00 | 1                                                     | -                                                     | $\begin{array}{c} 0.005 \pm \\ 0.0007452 \end{array}$ | 0.31                                    |
| 23.05.31 | 13:00~15:00 | $\begin{array}{c} 0.027 \pm \\ 0.0037224 \end{array}$ | $\begin{array}{c} 0.02 \pm \\ 0.0028152 \end{array}$  | -                                                     | 0.35                                    |
| 23.03.31 | 15:00~16:00 | -                                                     | -                                                     | $2.053 \pm 0.2828052$                                 | 0.35                                    |
| 23.06.08 | 11:00~12:00 | $\begin{array}{c} 0.007 \pm \\ 0.0009072 \end{array}$ | $\begin{array}{c} 0.015 \pm \\ 0.0021204 \end{array}$ | -                                                     | 0.42                                    |
| 23.06.23 | 11:00~12:00 | $\begin{array}{c} 0.043 \pm \\ 0.0055944 \end{array}$ | $\begin{array}{c} 0.002 \pm \\ 0.0001872 \end{array}$ | -                                                     | 0.42                                    |

#### 4. Conclusion

This study employed a mobile GHG measurement platform to quantify the methane emissions from a large LNG gas-fired power plant in Seoul, South Korea. Using our mobile measurement strategy, we identified significant fugitive methane sources at three hotspots (S1, S2, and S3) downwind of an LNG power plant. The results demonstrate the effectiveness of mobile measurement approaches for quantifying methane emissions from an urban LNG power plant and identifying specific emission hotspots in its vicinity. These findings highlight the need for targeted mitigation strategies, such as enhanced Leak Detection and Repair programs for pipelines and optimization of operational procedures (particularly during transient states, such as startup/shutdown) from LNG facilities in urban areas (Alvarez et al., 2018; Brandt et al., 2014; Howarth, 2019). However, because of restricted access to LNG power plants, we were unable to pinpoint the sources of methane emissions. Despite this limitation, our measurements confirmed that a substantial amount of methane was emitted from the LNG power plant. Direct measurements within a restricted area are required to obtain more accurate estimates of methane emissions.

Utilizing the GPDM and OTM-33A, we quantified methane emission rates associated with natural gas pipelines (S1: average  $0.09 \pm 0.0086$  tons  $hr^{-1}$ ; S2: average  $0.018 \pm 0.0015$  tons  $hr^{-1}$ ) and an exhaust pipe linked to internal facilities (S3: average  $0.55 \pm 0.0583$  tons  $hr^{-1}$ ). The maximum methane emission rate was quantified at the S3 hotspot, with a methane emission rate of  $2.053 \pm 0.283$  tons  $hr^{-1}$ . The top-down emission rates derived from our measurements showed significant discrepancies compared to the bottom-up inventory estimates, particularly during high-emission events. This disparity underscores the limitations of reliance on inventory methods that include fugitive methane emissions under various operating conditions.

| 353               | Author contributions                                                                                                                                                                                                                                |
|-------------------|-----------------------------------------------------------------------------------------------------------------------------------------------------------------------------------------------------------------------------------------------------|
| 354<br>355<br>356 | J. J and S. J designed and wrote the manuscript of this research. J. J, H. L, J. S, and D. K conducted mobile measurements. H. L, Y. L and D. C conducted data analysis for estimating methane emission rates.                                      |
| 357               |                                                                                                                                                                                                                                                     |
| 358               | Competing interests                                                                                                                                                                                                                                 |
| 359               | The authors declare no competing interests                                                                                                                                                                                                          |
| 360               |                                                                                                                                                                                                                                                     |
| 361               | Acknowledgements                                                                                                                                                                                                                                    |
| 362<br>363<br>364 | This work was supported by Korea Environment Industry & Technology Institute (KEITI) through Project for developing an observation-based GHG emissions geospatial information map, funded by Korea Ministry of Environment (MOE) (RS-2023-00232066) |
| 365<br>366        | Map data are the copyright of OpenStreetMap contributors and available from https://www.openstreetmap.org (last access: 18 October 2025)                                                                                                            |
| 367               |                                                                                                                                                                                                                                                     |
| 368               |                                                                                                                                                                                                                                                     |
| 369               |                                                                                                                                                                                                                                                     |

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
