# Peer review of "Methane quantification of LNG gas-fired power plant in Seoul, South Korea"

_EGUsphere, 2025_

## Referee Comment (RC1)

I reviewed the manuscript entitled *"Methane Quantification of an LNG Gas-Fired Power Plant in Seoul, South Korea"* with great interest. The study addresses an important knowledge gap by providing observational constraints on methane emissions in an understudied region. The authors conducted six months of mobile measurements and applied top-down (TD) approaches using a Gaussian Plume Dispersion Model (GPDM) and the OTM-33A method to quantify emissions. In addition, bottom-up (BU) estimates were used and compared with TD results. This integrated approach is valuable and timely.

However, while the study is promising, the manuscript requires substantial technical strengthening before it can be considered for publication. Several key methodological assumptions require clearer justification, the uncertainty analysis must be significantly expanded, and the comparison between TD and BU estimates needs stronger conceptual and quantitative support. Major and detailed comments are provided below.
* * *
**Major Comments**

The major concerns relate to the following aspects:

1) **Application of the GPDM and OTM-33A methods**
2) **Propagation and reporting of uncertainties**
3) **Comparison between TD and BU quantification; an inconsistency requires justification**
4) **Upscaling of short-term measurements to annual emissions**
5) **Attribution of detected emissions to specific industrial activities**
6) **Use of terminology, particularly "fugitive emissions"**
7) **Consideration of financial losses associated with methane emissions**

(a) Points 1 & 2: Applicability and Uncertainty of GPDM and OTM-33A

The Gaussian plume dispersion method relies on several strict assumptions, including flat terrain, homogeneous wind fields, steady-state atmospheric conditions, and a single well-defined point source. For OTM-33A, a consistent wind direction within ±30°, ideally for more than 80% of the sampling period, is additionally required. Measurement distance from the source is also a critical parameter.

The manuscript does not adequately specify:

- The extent to which these assumptions were met,

- How deviations from these conditions may have influenced the emission estimates,

- How associated uncertainties were propagated into the final results.

The authors must explicitly report all measured parameters that influenced the final quantification (e.g., wind speed, wind direction variability, source-receptor distance, stability class) and quantitatively assess their contributions to total uncertainty.

(b) Points 3 & 4: BU–TD Comparison and Annual Upscaling

The BU approach is inherently designed for annual emission estimation, whereas the TD approach in this study reflects snapshots during limited sampling periods. It appears that the BU estimates were applied to the temporal window of the measurements, rather than being strictly annualized.

The manuscript must clearly explain:

- How the TD annualized estimates were consistently compared with BU estimates.
  From Table 3, inventory values are, in several cases, greater than the measured TD values for S1, S2, and in three out of five cases for S3. This contradicts the authors' conclusion that inventories underestimate emissions. The data currently presented suggest the opposite. This inconsistency must be addressed. Additionally, annual variability in emissions must be discussed, as it directly affects the validity of BU–TD comparisons.
- How short-term TD emission estimates were upscaled to annual values,
- Whether temporal variability (seasonal, operational, maintenance-related) was accounted for,

**(c) Point 5: Source Attribution**

The manuscript provides insufficient methodological detail on how emissions were attributed to specific industrial activities. This attribution is central to the interpretation of results and to the applicability of BU estimates

**(e) Point 7: Financial Impacts of Methane Loss**

Beyond environmental and health impacts, methane loss also represents a direct economic loss, particularly for LNG-importing countries. The manuscript would be significantly strengthened by:

- Estimating annual financial losses associated with both intentional and unintentional methane emissions,

- Clearly separating fuel loss mechanisms,

- Discussing implications for national energy security and policy.
* * *
**Detailed Comments**

**Abstract**

The abstract should be more specific and quantitative, highlighting key numerical findings and methodological outcomes rather than relying on general statements.

**Introduction**

Additional literature review is required to better contextualize methane emissions from industrial sites. In the result and discussion you can compare quantifications of methane emissions from the LNG site with other sources in urban area of Seoul.

**Materials and Methods**

- Key methodological steps require substantial clarification and expansion.

- The background subtraction method is not described and must be explicitly explained.

- L126: Frequency and protocol of instrument calibration must be reported.

- L158–169: Explicitly describe operational requirements and data screening criteria for GPDM and OTM-33A.

- L172: Clarify how atmospheric stability classes were determined.

- L185: Allocating monthly consumption to 5-min intervals does not capture startup and shutdown behavior; discuss limitations and potential bias.

- L188: Justify the use of post-meter leakage emission factors for exhaust-based emissions.

- L194: Replace with "site access restrictions" for clarity.

- L205: Clarify whether this refers to the surrounding area of the power plant.

- L225–228: This section should be moved to Methods as a subsection on source attribution strategy.

**Study Area Description**

- L113–117: If available, provide detailed information on:

  o LNG storage capacity,

  o Reported emissions from storage systems,

  o Number and type of exhausts,

  o Surrounding infrastructure affecting plume behavior.

- L129: GPS precision of ±20 m is coarse for GPDM applications; clarify whether this refers to horizontal coordinates or elevation.

**Results and Discussion**

Further clarification is required regarding:

1. Uncertainty ranges and their derivation,

2. Annual upscaling and consistency with BU estimates,

3. Interpretation of TD–BU discrepancies,

4. Robust identification of emission sources,

5. Estimation of financial losses.

- L18–20: See major comment on source attribution.

- L20–21: Revise significant digits, clearly distinguish whether values are instantaneous or annualized.

- L24–25: Place the root causes of TD–BU discrepancies here.

- L26–27: Justify why mobile measurements were particularly advantageous in the Seoul context.

- L21–23: Provide deeper explanation of emission processes for S1–S3, especially S3.

- L25: Explicitly discuss limitations of BU approaches.

- L35–49: Overly long; tighten to focus on manuscript objectives.

- L48: Check citation formatting.

- L58–61: Clarify that uncertainties remain large for LNG-specific methane emissions.

- L80–81: Later use these values for cross-site comparisons.

- L211–213: Avoid excessive decimal precision.

- L214: Methane enhancement alone does not prove source strength.

- L233 onward: Reduce decimals; explain how such small uncertainty ranges were derived.

- L247–250: Root-cause attribution appears speculative; consider intentional venting.

- L251–272: Separate BU–TD comparison from variability discussion.

- L268: Super-emitters must be defined consistently; include dispersed source contributions (Williams et al., 2025).

- L289: Compare S3 with other documented super-emitting sites using flux-based metrics.

- L332–334: Focus conclusions on study results, not on method validation alone.

- L339: Emphasize transparency issues.

- L343: Expand on temporal variability and its implications.

- L347–351: Discuss whether GPDM and OTM-33A assumptions were truly met in this complex environment.
* * *
**Figures and Tables**

**Figures**

- Figure 5a is blurry.

- Axis labels in Figures 5b and 5c are unreadable; consider using ppm units.

**Tables**

- Table 3: It is unclear whether inventory values apply to S1, S2, or combined sources. Inventory values exceed measured values in several cases; this must be clarified and reconciled with conclusions